# BrainMIND: Interpret Fine-grained Spatial Mapping of Brain Activity to Multi-semantic Concepts

## Abstract

Understanding how population-coding in the human visual cortex shape high-level semantic representations remains a significant challenge. Prior work has either focused on region-level text decoding or relied on simple linear models to probe single-semantic decoding at the voxel level. Consequently, systematic exploration of semantic diversity remains limited at both the region level and the fine-grained voxel level. To address this gap, we introduce BrainMIND, a data-driven framework for analyzing multi-concept semantic selectivity in the visual cortex. We use a conditional variational autoencoder (CVAE) whose latent space is constrained by brain data and spatial locations of voxels. The CVAE decodes the structured latent space into CLIP-aligned semantic embeddings, which then condition a fine-tuned large language model to generate interpretable captions. We validate Brain-MIND on widely recognized cortical regions, demonstrating interpretable region-level and voxel-level semantic selectivity. We reveal that individual voxels exhibit mixed selectivity across multiple semantic dimensions, and filling a key gap in voxel-wise neural decoding. Our results demonstrate that BrainMIND provides an interpretable bridge from brain regions to their constituent voxels, enabling controlled, fine-grained exploration of semantic organization in the higher visual cortex.

## 1 Introduction

Decoding the functional architecture of the brain remains a central challenge in neuroscience (Raichle, 2009). A key aspect of this domain is understanding how the human brain processes complex visual information, which relies on a network of specialized high-level visual areas (Kanwisher et al., 1997; Grill-Spector, 2003). Prior foundational research successfully identified cortical regions with strong selectivity for specific semantic categories, such as faces, places, words, and bodies (Puce et al., 1996; Kanwisher et al., 1997; Epstein & Kanwisher, 1998; Downing et al., 2001; Grill-Spector, 2003). However, these hypothesis-driven approach, which relies on manually selected stimuli, may underestimate the richness of real-world images, thereby limiting our understanding of fine-grained, voxel-level resolution and causing us to overlook how a single brain region or voxel can encode multiple, overlapping semantic concepts. (Huth et al., 2012).

Recent work has begun leveraging high-level semantic features extracted from contrastive vision–language models (e.g., CLIP) with large generative models to perform end-to-end reconstruction of fMRI activity. These methods typically first map brain signals into CLIP's latent space, and then synthesize images or textual descriptions using diffusion models or generative adversarial networks, thereby revealing how the brain represents natural scenes (Takagi & Nishimoto, 2023; Chen et al., 2023; Ozcelik & VanRullen, 2023; Ferrante et al., 2023; Liu et al., 2023; Gu et al., 2022; Luo et al., 2023a). Despite notable gains in reconstruction quality, these systems are primarily optimized for generating naturalistic stimuli rather than for fine-grained interpretation of neural representations.

Complementarily, voxel-wise encoding models probe the relationship between semantic features and neural activity. These approaches have achieved significant success in the visual domain, for example, by evaluating the predictive power of features from models like CLIP on fMRI signals in

the ventral visual pathway (Radford et al., 2021; Naselaris et al., 2011; Wen et al., 2018a). Extensions to language decoding either use regression with CLIP features on visual data and transfer the learned weights to text (Luo et al., 2023b), or directly leverage large language models to decode language-related brain activity (Qiu et al., 2025). Although progress has been made, current approaches tend to overlook how specific regions or voxels encode and integrate multiple, overlapping semantic concepts, and thus still lack a fine-grained account.

To better address these issues,we introduce **BrainMIND** (Multi-semantic Interpretable Neural Decoding), a novel framework to investigate multi-semantic selectivity from brain activity at both the Region of Interest (ROI) level and from large populations of individual voxels. BrainMIND is trained on the Natural Scenes Dataset (NSD) (Allen et al., 2022), a massive-scale collection of fMRI responses paired with visual stimuli. In our methodology, images from the NSD are first encoded into vision-language aligned embeddings using the CLIP model.

We propose a method that maps CLIP features to a latent space modeled by a target Gaussian Mixture Model (GMM). The mapping is trained to project CLIP embeddings that maximally activate a voxel into low-variance regions of the GMM. Critically, a single, unified model jointly encodes the preferred visual representations for all voxels by conditioning on each voxel's spatial position, which dramatically improves training efficiency.

By obtaining semantic interpretations for both an ROI and all of its constituent voxels, BrainMIND enables the generation of a fine-grained, multi-semantic spectrum of the higher visual cortex. Furthermore, our framework facilitates multi-level explorations of the relationships between the semantics encoded in different voxels, as well as between an ROI and its underlying voxel population.

We list our contributions below:

- Our framework achieves multi-semantic and position-aware decoding at both the coarse ROI scale and the fine-grained voxel level, providing a comprehensive spatial view of neural representation.

- We introduce a scalable nonlinear decoding pipeline, the semantic representations decoded by our framework exhibit superior alignment with established neuroscientific priors and enhance the interpretability of concepts identified by antecedent methods.

- We reveal the intricate semantic correspondence between individual voxels within a brain region and across regions. And we also find the semantic relevance of fine-grainded voxels do not perfectly align with their physical location proximity.

## 2 RELATED WORK

**Brain semantic decoding in higher visual cortex.** The human higher visual cortex contains regions selective for semantic categories like faces, places, and bodies, traditionally identified using handcrafted stimuli (Puce et al., 1996; Kanwisher et al., 1997; Epstein & Kanwisher, 1998; Downing et al., 2001; Grill-Spector, 2003). However, this approach may poorly capture neural tuning under naturalistic conditions (Felsen & Dan, 2005), leading to a shift toward data-driven methods. Modern voxel-wise encoding models now map features from natural stimuli like movies to brain responses (Naselaris et al., 2011; Huth et al., 2012). In particular, deep neural networks have proven effective for building these predictive models, allowing for data-driven exploration of the visual hierarchy (Yamins et al., 2014; Wen et al., 2018b).

**Caption and image generation conditioned on brain activation.** Recent research has advanced from simple classification to generating rich outputs like images and text directly from brain activity. Early efforts in visual reconstruction employed generative adversarial networks (GANs) to synthesize images from fMRI signals (Shen et al., 2019; Seeliger et al., 2018). More recently, diffusion models have enabled high-fidelity image reconstructions that closely resemble perceived stimuli (Takagi & Nishimoto, 2023; Ozcelik & VanRullen, 2023; Chen et al., 2023). Parallel work generates natural language captions by mapping brain activity to language model embeddings, thus describing the semantic content of neural representations (Luo et al., 2023b; Défossez et al., 2023). Beyond pure decoding, generative models now serve as tools for scientific inquiry. Methods such as BrainACTIV and BrainDiVE use diffusion models to manipulate or synthesize stimuli, respectively, to efficiently map the tuning properties of cortical regions (Luo et al., 2023a). These generative ap-

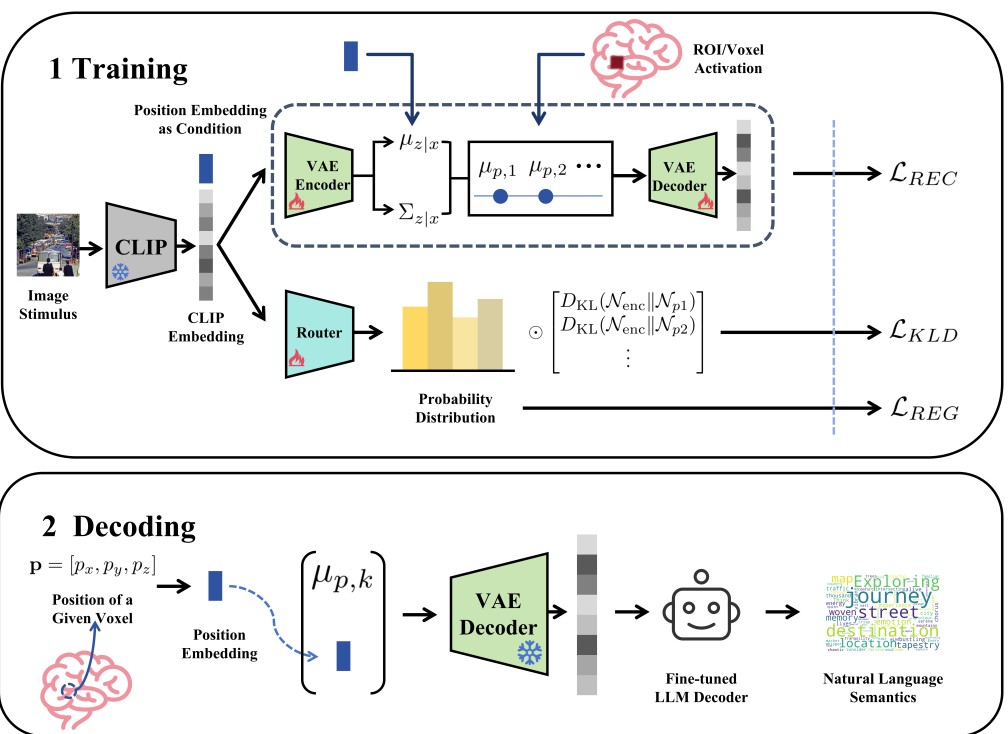

Figure 1: **Pipeline of our BrainMIND.** (1) **Training stage:** A voxel's $(p_x, p_y, p_z)$ position is encoded into a position embedding, concatenated with the stimulus's CLIP embedding, and passed into a CVAE encoder. The resulting latent representation is then regularized by a dynamic prior—informed by fMRI data—which consists of K distributions weighted by a router module. (2) **Semantic decoding stage:** To decode the $K$ semantic concepts for a given voxel, its position embedding $\mathbf{e}_p$ is first concatenated with each of the $K$ latent centers $\boldsymbol{\mu}_{p,k}$. These combined vectors are then passed through the CVAE Decoder to generate semantic embeddings ($\hat{\mathbf{x}}_k$), which are finally translated into natural language descriptions by a fine-tuned LLM.

proaches rely on powerful encoding models that accurately map visual features to brain responses, often leveraging the rich representations of vision-language models (Yamins et al., 2014; Wang et al., 2023).

In summarize, prior work has made notable progress in identifying brain function at the ROI level and in semantic decoding at the voxel level. Nevertheless, these approaches are limited as they cannot perform multi-semantic decoding for individual ROIs or voxels. Moreover, methods that depend on linear regression of fMRI signals often exhibit suboptimal performance, typically manifested by low test correlations(Benara et al., 2024). Our work concentrates on achieving versatile multi-semantic decoding at both the ROI and voxel scales. We enable the simultaneous application of non-linear techniques for multi-semantic decoding across thousands of voxels, addressing the limitations of previous linear models.

## 3 METHODOLOGY

### 3.1 PROBLEM SET

Our goal is to model the semantic tuning of brain regions and voxels by analyzing their fMRI responses ($\mathcal{R}$) to a set of visual stimuli ($\mathcal{I}$). From the collected stimulus-response pairs, $\{(i_n, r_n)\}_{n=1}^{N}$, we aim to determine the semantic tuning of each brain region and voxel in the higher-level visual cortex. To represent the images in a semantically rich feature space and reduce noise, we use the pre-trained CLIP image encoder to extract a feature vector $\mathbf{x} \in \mathbb{R}^{D_{in}}$ for each image $i$: $\mathbf{x} = \text{CLIP}(i)$.

## 3.2 CONDITIONAL VAE WITH A DYNAMIC MIXTURE PRIOR

We propose a model built upon a Conditional Variational Autoencoder (CVAE) framework. This model is uniquely enhanced with a dynamic mixture-of-Gaussians prior, where the prior's structure is directly modulated by the observed fMRI responses.

### 3.2.1 CONDITIONAL VARIATIONAL AUTOENCODER

The core of our model is a Conditional Variational Autoencoder (CVAE) (Kingma & Welling, 2013; Sohn et al., 2015), designed to reconstruct a CLIP image feature $\mathbf{x}$ conditioned on a given 3D voxel position $\mathbf{p}$. We first map the physical coordinates $\mathbf{p}$ of the voxel to a high-dimensional positional embedding $\mathbf{e}_p$ via a position encoder, which serves as the central condition for the model. During the encoding phase, this positional embedding is concatenated with the input feature, as $[\mathbf{x}; \mathbf{e}_p]$, and fed into the encoder to infer the latent variable's distribution. Similarly, in the decoding phase, a latent variable $\mathbf{z}$ sampled from this distribution is again concatenated with the same positional embedding $\mathbf{e}_p$ and passed to the decoder to perform the feature reconstruction. By explicitly injecting this positional condition into both the encoder and decoder, our model learns to capture structured, spatially-dependent variations in the features.

### 3.2.2 DYNAMIC MIXTURE PRIOR VIA ROUTING

Instead of a standard static prior (e.g., $\mathcal{N}(\mathbf{0}, \mathbf{I})$), we introduce a dynamic mixture-of-Gaussians prior to capture the multi-semantic structure of neural representations. The selection of the appropriate mixture component for a given clip embedding input $\mathbf{x}$ is handled by a **Router** network, parameterized by $\psi$.

**Router Network.** The Router takes the CLIP feature $\mathbf{x}$ and outputs a probability distribution $\boldsymbol{\pi}$ over $K$ latent prior components:

$$\boldsymbol{\pi}(\mathbf{x}) = \{\pi_1(\mathbf{x}), \ldots, \pi_K(\mathbf{x})\}$$
$$= \mathrm{Softmax}(\mathrm{NN}_\psi(\mathbf{x})), \qquad \sum_{k=1}^{K} \pi_k(\mathbf{x}) = 1. \tag{1}$$

Each $\pi_k(\mathbf{x})$ can be interpreted as the probability that the stimulus represented by $\mathbf{x}$ belongs to the $k$-th latent semantic cluster.

**Dynamic Prior Formulation.** The prior distribution over $\mathbf{z}$ is a mixture model. Its component parameters are dynamically determined by the fMRI brain response $r \in \mathbb{R}$, corresponding to a specific ROI or voxel, and a scaling hyperparameter $\tau$. The prior for a given input $\mathbf{x}$ and its corresponding response $r$ is:

$$p(\mathbf{z}|\mathbf{x}, r, \tau) = \sum_{k=1}^{K} \pi_k(\mathbf{x}) \mathcal{N}(\mathbf{z}|\boldsymbol{\mu}_{p,k}(r, \tau), \boldsymbol{\sigma}_p^2(r, \tau)\mathbf{I}) \tag{2}$$

The parameters are defined as:

$$\boldsymbol{\mu}_{p,k}(r, \tau) = \mathrm{sign}(r) \cdot (1.0 + k) \cdot \mathbf{1} \tag{3}$$
$$\boldsymbol{\sigma}_p^2(r, \tau) = |\frac{\tau}{r}| \tag{4}$$

where $\mathbf{1}$ is a vector of ones of dimension $D_z$. This formulation establishes a crucial inverse relationship between the prior's variance and the measured brain activity. A strong fMRI response $r$ indicating high activation of the ROI or voxel for the given input, results in a smaller prior variance $\boldsymbol{\sigma}_p^2$. Note that $r$ is data and $\tau$ is a hyperparameter instead of training parameters. This concentrates the latent representation $\mathbf{z}$ more tightly around one of the selected prior means. Consequently, at the $K$ center positions of the Gaussian distributions($\boldsymbol{\mu}_{p,k} = (1.0 + k) \cdot \mathbf{1}$), which correspond to peak activation levels where $r$ is maximized, the prior variance $\boldsymbol{\sigma}_p^2$ theoretically approaches zero. This enforces a highly structured and confident latent encoding for stimuli that elicit the strongest neural responses.

### 3.2.3 OBJECTIVE FUNCTION

The model is trained end-to-end by minimizing a total loss function $\mathcal{L}_{\text{total}}$, which is a weighted sum of three distinct components: a reconstruction loss, a Kullback-Leibler (KL) divergence loss, and a router regularization term.

$$\mathcal{L}_{\text{total}} = \omega_{\text{recon}}\mathcal{L}_{\text{recon}} + \omega_{\text{KL}}\mathcal{L}_{\text{KL}} + \omega_{\text{router}}\mathcal{L}_{\text{router}} \tag{5}$$

**Reconstruction Loss.** The reconstruction loss encourages the CVAE to faithfully reproduce its input, which in our case is the CLIP feature vector. We use the mean squared error (MSE) between the original feature $\mathbf{x}$ and its reconstruction $\hat{\mathbf{x}}$:

$$\mathcal{L}_{\text{recon}} = \|\mathbf{x} - \hat{\mathbf{x}}\|_2^2 \tag{6}$$

**KL Divergence Loss.** The KL divergence term regularizes the latent space. It measures the divergence between the encoder's approximate posterior $q_\phi(\mathbf{z}|\mathbf{x}, \mathbf{p})$ and our dynamic mixture prior $p(\mathbf{z}|\mathbf{x}, r, \tau)$. The loss is the expected KL divergence over the mixture components, weighted by the router probabilities:

$$\mathcal{L}_{\text{KL}} = \sum_{k=1}^{K} \pi_k(\mathbf{x}) D_{KL}\left(q_\phi(\mathbf{z}|\mathbf{x}, \mathbf{p}) \,\|\, \mathcal{N}(\mathbf{z}|\boldsymbol{\mu}_{p,k}(r, \tau), \boldsymbol{\sigma}_p^2(r, \tau)\mathbf{I})\right) \tag{7}$$

**Router L2 Regularization.** To prevent the router from collapsing into a state where it produces one-hot like probability distributions for all inputs, we apply an L2 penalty to its output probability vector:

$$\mathcal{L}_{\text{router}} = \|\boldsymbol{\pi}(\mathbf{x})\|_2^2 = \sum_{k=1}^{K} \pi_k(\mathbf{x})^2 \tag{8}$$

### 3.3 LLM-BASED TEXT GENERATION

For any given voxel at position $\mathbf{p}$, we can interpret its learned semantic tuning by converting its core representational concepts into natural language:

We generate the canonical CLIP embedding for each of the $K$ latent semantic concepts that the voxel is tuned to. We begin by obtaining the voxel's position embedding $\mathbf{e}_p$ using our trained position encoder. Then, for each of the $K$ learned prior means $\boldsymbol{\mu}_{p,k}$ (which represent the centers of the latent semantic clusters), we use the trained CVAE decoder to reconstruct the clip embedding $\hat{\mathbf{x}}_k$:

$$\hat{\mathbf{x}}_k = \text{Decoder}([\boldsymbol{\mu}_{p,k}; \mathbf{e}_p]), \quad \text{for } k = 1, \ldots, K. \tag{9}$$

Each resulting vector $\hat{\mathbf{x}}_k$ serves as a clip representation of a core semantic concept encoded by the voxel at position $\mathbf{p}$.

Second, these generated CLIP embeddings $\{\hat{\mathbf{x}}_k\}_{k=1}^{K}$ are translated into descriptive text by a fine-tuned LLM.

### 3.3.1 LLM ADAPTATION AND OBJECTIVE

Given a semantic vector $\hat{\mathbf{x}}_k \in \mathbb{R}^{d_c}$, we obtain a length-$n$ *visual prefix* via a lightweight projector $P_\phi$: $\mathbf{V}_k = P_\phi(\hat{\mathbf{x}}_k) \in \mathbb{R}^{n \times H}$ (projector details in the Appendix). The backbone LLM remains frozen; we train only (i) the projector parameters $\phi$ and (ii) a small set of LoRA adapters on attention layers. Let $E_{\text{text}} \in \mathbb{R}^{T \times H}$ denote the embedding sequence of the textual prompt. We concatenate the visual prefix and the prompt embeddings along the sequence dimension and feed the result via `inputs_embeds`:

$$E_{\text{in}} = \text{concat}(\mathbf{V}_k, E_{\text{text}}), \tag{10}$$

where $n$ is the prefix length and $H$ is the LLM hidden size. Training minimizes the standard next-token cross-entropy on the target caption $y_i = (y_{i,1}, \ldots, y_{i,T})$:

$$\mathcal{L} = -\sum_{t=1}^{T} \log p_\Theta(y_{i,t} \mid E_{\text{in}}, y_{i,<t}), \tag{11}$$

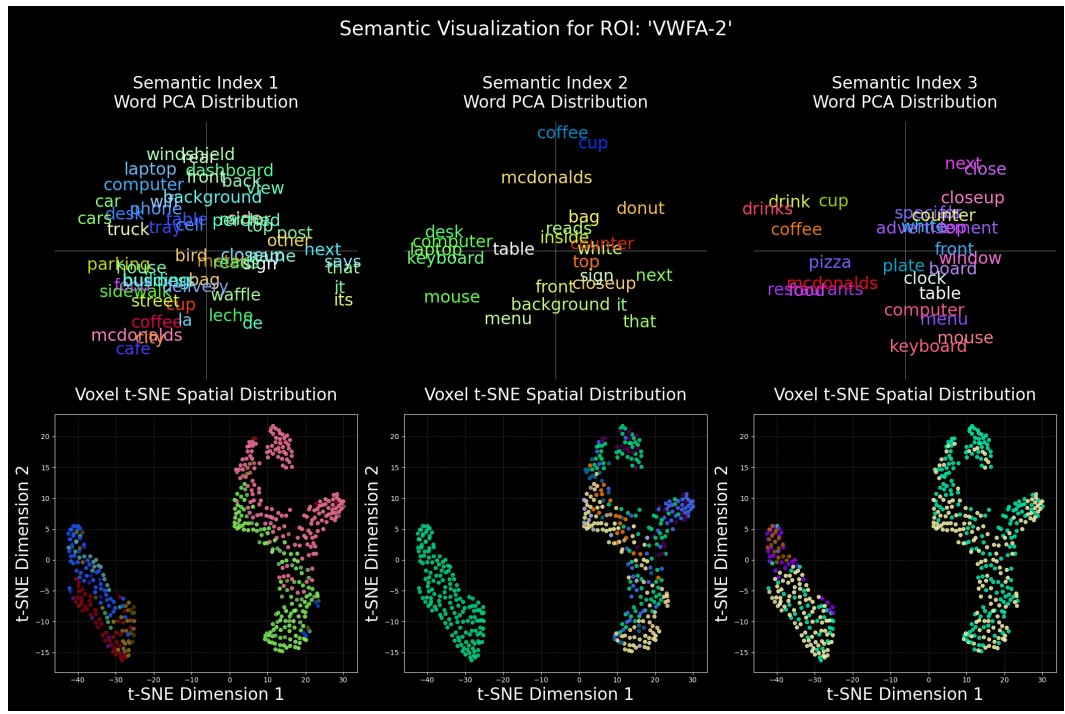

Figure 2: **Words Generated by BrainMIND** (first three semantics of all OVWFA's voxels, for more can refer to Appendix). For each region, we decoded words from all constituent voxels, selecting their principal semantic representation. To visualize the semantic space, we generated word embeddings using MiniLM-L6-v2 and projected them into a 3D color space using PCA. In the lower panel, we visualize the spatial relationships of the source voxels by applying t-SNE to their 3D coordinates to obtain a 2D topological map. The voxels are colored in correspondence with their decoded word's semantic embedding. This visualization reveals a strong clustering effect among voxels belonging to the same ROI.

where $\Theta$ contains the projector and LoRA parameters only. At inference, we compute $\mathbf{V}_k$ with $P_\phi$, form $E_{\text{in}}$ as in equation 10, and call `generate(inputs_embeds=` $E_{\text{in}}$`)` for autoregressive decoding; deterministic decoding is the default, with standard sampling strategies optionally enabled for diversity.

## 4 RESULTS

### 4.1 EXPERIMENT SETUP AND EVALUATION

All experiments were conducted on the Natural Scenes Dataset (NSD) Allen et al. (2022), a large-scale, high-resolution 7T fMRI dataset. The dataset contains brain responses from subjects viewing thousands of natural scenes from the COCO dataset. For our analysis, we performed a $z$-score normalization on the voxel response data for each experimental session independently. This within-session normalization ensures that the data from each session has a mean of zero and a standard deviation of one.

For the pretrained models, we employed include **CLIP-ViT-H-14** as the CLIP model,**Vicuna-7B** as the LLM, and **Stable Diffusion v1.5** as the diffusion model. For the training details of our model, please refer to the Appendix C.1.

To evaluate the decoding accuracy of BrainMIND at the voxel level, we identified the k most frequent nouns within the text generated for a given brain region. We then computed the cosine similarity between these nouns and their corresponding high-level concepts (i.e., faces, words, places, bodies),

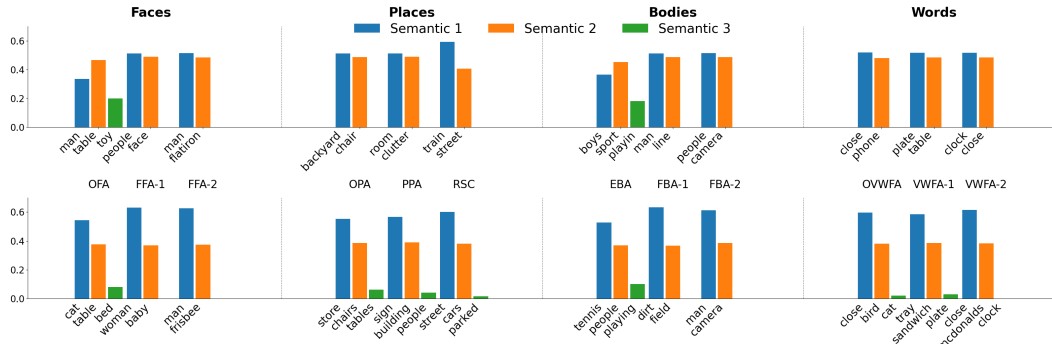

Figure 3: **Distribution of four semantic components at the ROI level (top) and the voxel level (bottom).** It is evident that the first two semantics are predominant. For K=4, the fourth semantic component is absent across all ROIs and voxels. A comparison of the two levels indicates that multi-semantic selectivity is slightly more pronounced for individual voxels than for entire ROIs.

using MiniLM-L6-v2 as the text encoder. As a baseline, we performed the same analysis on the top-k nouns decoded by BrainSCUBA (Table 4.1).

The comparison reveals that BrainMIND generates text more consistent with established neural priors for these regions, thereby confirming the model's validity and effectiveness.

| Method | Concept | | | |
|---|---|---|---|---|
| | **Words** | **Places** | **Faces** | **Bodies** |
| BrainSCUBA | 0.298 | 0.299 | 0.365 | 0.357 |
| Ours | **0.321** | **0.308** | **0.404** | **0.368** |

Table 1: **Semantic Similarity Comparison.** Comparison of cosine similarity between the top 10 decoded words and the corresponding ground-truth concept. We compare our method against the BrainSCUBA baseline.

## 4.2 REGION-LEVEL SEMANTIC DECODING

**Different ROIs focus on different concepts.** From Figure 3, region-level decodes align with the expected specializations. The **Place** ROI yields a concise, scene-centric summary (e.g., "bed,painting, laptop, table"), emphasizing indoor layout. The **FFA** ROI produces a clear people/face-centric description (e.g., "people,men, women, face"). The **Body** (EBA/FBA-like) ROI highlights human/body/action content (e.g.,"leg,hand, women, sports"). In short, each ROI's overall decode captures its dominant semantic tendency—*scene* for Place, *people/faces* for FFA, and *body/action* for Body.

**ROIS respond to multi-selective phenomenon.** To investigate how ROIs respond to natural visual stimuli, we randomly selected 5000 natural images. We set $K = 4$ in our model and used its router to obtain the components of each image's CLIP embedding across four Gaussian distributions. The results are shown in Figure 3. The results indicate that relatively early visual areas, such as OPA, OFA, EBA, and OVWFA, exhibit stronger mixed selectivity.

## 4.3 VOXEL-LEVEL SEMANTIC DECODING

**Voxels show clustered activation patterns.** In Section 4.2, we validate our method by decoding at the ROI level : the results demonstrate that brain regions associated with different concepts (i.e., faces, words, bodies, and places) exhibit distinct selectivity, which is consistent with established neuroscience priors. Furthermore, our method reveals that within a single ROI, voxels form semantic clusters (see Figure 2). Generally, physically adjacent voxels tend to exhibit similar semantic properties; however, they can also represent disparate semantics. For instance, as illustrated in Fig-

| Distance cluster | Voxel A $(p_x, p_y, p_z)$ | Voxel B $(p_x, p_y, p_z)$ | Euclidean Distance | Text Decoding |
|---|---|---|---|---|
| Near | [60, 37, 34] | [60, 37, 35] | 1.00 | two men standing next to each other with *one man holding a camera.* |
| Near | [59, 38, 28] | [59, 38, 29] | 1.00 | **A:** a man holding a camera next to a man in a wheelchair. **B:** two men standing next to each other with one holding a cell phone. |
| Far | [59, 35, 32] | [61, 45, 33] | 10.25 | two men standing next to each other with *one man holding a camera.* |
| Far | [59, 31, 34] | [61, 47, 30] | 16.61 | **A:** a man holding a cell phone in front of a mirror in a store. **B:** a man holding a camera next to a man in a wheelchair with a dog |

Table 2: FFA-2 voxels and LLM-decoded outputs grouped by spatial separation. Rows 1–2 and 3–4 illustrate voxel pairs with small Euclidean distances that nevertheless produce different decoded sentences. In contrast, the voxel in first line and the third line are far apart in Euclidean distance yet yield the same decoded sentence. Taken together, these cases show that within-cluster consistency in the decoded semantic space is not totally explained by spatial proximity.

ure 2, for the first semantic category within VWFA-2, voxels with x-coordinates in the range of [0, 30] are physically clustered, yet they are decoded into two distinct semantic classes.

**Voxels respond to diverse semantic stimuli.** Following a similar methodology to Section 4.2, we analyzed the router outputs for a large corpus of natural images (Figure 3, bottom panel). The analysis reveals that mixed selectivity is more pronounced at the voxel level than at the ROI level, particularly within the 'places' and 'words' regions. For instance, within the 'words' processing hierarchy, selectivity for the third semantic component progressively diminishes from OVWFA, to VWFA-1, and subsequently to VWFA-2. This trend is consistent with prior findings showing that posterior occipitotemporal regions (e.g., OVWFA) exhibit broader, multi-component tuning, whereas more anterior word-selective regions (VWFA-1 and VWFA-2) become increasingly specialized and lexically tuned (Vinckier et al., 2007; Lerma-Usabiaga et al., 2018; White et al., 2019).

| Voxel $(p_x, p_y, p_z)$ | Decoded sentences |
|---|---|
| [17, 18, 37] | 1. close up plate food tray. 2. close sandwich plate napkin top table. 3. slices toast sandwich top 4. plate food tray top table. |

Table 3: An example of single-voxel from VWFA-1, showing four unique sentences from the same voxel, demonstrating that one voxel can support multiple semantic readouts (mixed selectivity).

**Effect of voxel spatial proximity on neural decoding consistency.** To investigate whether the spatial proximity of voxels affects the consistency of neural decoding, we analyzed pairs of voxels within specific ROIs. We quantified the physical distance and the decoded semantic similarity for each voxel pair $(i, j)$. Specifically, physical distance was calculated as the Euclidean norm of their 3D coordinates, $\|\mathbf{p_i} - \mathbf{p_j}\|_2$, while semantic similarity was measured by the cosine similarity of their decoded semantic vectors(here we take the first of all $K$ semantics as a example), $(\mathbf{s_i}, \mathbf{s_j})$, $sim(\mathbf{s_i}, \mathbf{s_j}) = \frac{\mathbf{s_i} \cdot \mathbf{s_j}}{\|\mathbf{s_i}\| \cdot \|\mathbf{s_j}\|}$.

As Figure 4 illustrates, from an overall perspective, a distinct negative correlation is clearly observable: as the physical distance between voxels increases, the similarity of their decoded content tends to decrease. A linear regression analysis (red dashed line) quantifies this macro-level trend, yielding a strong coefficient of determination ($R^2 = 0.885$). However, at a more fine-grained level, the plot reveals that for any given physical distance(for instance, at a distance of 10), there is a substantial margin and high variance in similarity values. This indicates that even physically proximate voxels can generate semantically diverse content.

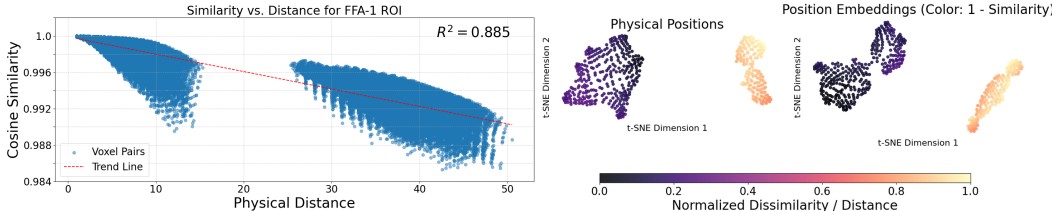

Figure 4: **(a) Left:** Decoded semantic similarity vs. physical distance for voxels within the FFA-1 ROI($R^2 = 0.885$). **(b) Right:** t-SNE embeddings of two representations for all voxels within the FFA-1 region: (left) the raw physical coordinates, and (right) the model-encoded position embeddings.

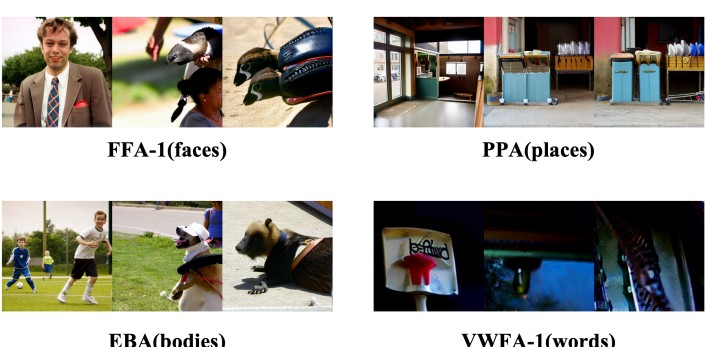

Figure 5: Images generated by diffusion model with utilizing the multi-semantic embeddings generated from BrainMIND. Results demonstrated preferred images from FFA-1, PPA, EBA, VWFA-1.

We also investigate how semantic context modulates the topological structure of the position embeddings learned by our model. The results (The right part of Figure 4) show that while the global topology of the embeddings is largely congruent with the physical voxel space, we observe subtle but distinct discrepancies at a fine-grained level, where embedded positions do not perfectly align with their relative physical locations.

### 4.4 CLIP-GUIDED DIFFUSION IMAGE GENERATION

As shown in Figure 5, we use a diffusion model (Ho et al., 2020; Ye et al., 2023) to visualize our multi-semantic embeddings. We generate images based on the multi-semantic concepts from BrainMIND decoded from various ROIs. For each brain region presented, the image generated from the primary semantic concept is highly consistent with established neuroscientific priors. In contrast, the two remaining images for each region exhibit significant semantic diversity.

## 5 DISCUSSION AND LIMITATIONS

**Conclusion.** In this paper, we introduced BrainMIND, a novel data-driven methodology for decoding multi-semantic concepts from brain activity at both roi and fine-grained voxel level. Our results yield several key insights into neural coding. We found that a significant portion of brain ROIs and their constituent voxels exhibit mixed selectivity, indicating that individual neural units are tuned to multiple, varied semantic concepts rather than just one. Furthermore, our analysis revealed a quantifiable relationship between the functional similarity of voxels and their physical proximity, as well as distinct clustering patterns among voxels within unified ROIs. Collectively, these findings suggest that BrainMIND is a powerful tool capable of revealing fundamental principles of neural organization and can serve as a valuable asset for advancing neuroscientific research.

**Limitations and Future Work.** we acknowledge two primary limitations in the current BrainMIND framework. First, the text-decoding component is fundamentally dependent on a pre-trained Large

Language Model (LLM). Consequently, the decoded semantics may inherit intrinsic biases from the underlying LLM, which fine-tuning may not fully eradicate. Second, a more foundational challenge in brain decoding is the absence of a definitive ground truth. Therefore, the evaluation of our model has to rely on indirect validation methods. Future work will directly address these challenges. To mitigate the risks of LLM-induced biases, we plan to explore decoding architectures that incorporate factuality constraints and verify ground truths for specific neural phenomena, enabling a more rigorous assessment of decoding accuracy.

## 6 ETHICS STATEMENT

Our study uses only publicly released, de-identified human fMRI data from the Natural Scenes Dataset (NSD). we performed no new data collection and did not attempt any re-identification. We caution against any consequential use (e.g., clinical, legal, or employment decisions) and discuss technical and conceptual limitations in the paper. Because our pipeline relies on CLIP-aligned features and large language models, which can inherit social and representational biases, we (i) avoid demographic inferences, (ii) report failure cases, and (iii) include guardrails to prevent unsafe or sensitive outputs. We release only code, trained weights, and derived embeddings permissible under the NSD data-use terms; no raw or identifiable human data are shared.

## 7 REPRODUCIBILITY STATEMENT

We will release the full codebase, pretrained checkpoints, and step-by-step scripts to reproduce all results at our GitHub repository. More implementation details are included in Appendix.

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

APPENDIX

## A    LLM USAGE STATEMENT

In preparing this manuscript, we used ChatGPT and Gemini strictly as a writing assistant for grammar and clarity on author-written passages (e.g., wording tweaks in the Abstract/Introduction). No sections were drafted by the LLM, and the model was not used for research ideation, dataset construction, data preprocessing/analysis, model design, figure generation, experiments, or citation retrieval. We did not input any confidential or non-public human data beyond the manuscript text. All scientific content, claims, and references were produced and validated by the authors, who take full responsibility for the paper's contents, and the use of the LLM does not imply authorship.

## B    ALGORITHM BOX

---

**Algorithm 1** Training the Conditional VAE with Router for Voxel Representation

---

**Require:** Training data $\mathcal{D} = \{(\boldsymbol{p}_i, \beta_i, \boldsymbol{x}_i)\}_{i=1}^{N}$, where $\boldsymbol{p}$ is voxel position, $\beta$ is fMRI activity, and $\boldsymbol{x}$ is image embedding.
**Require:** Model parameters $\theta$ (CVAE) and $\phi$ (Router), number of mixture components $K$, loss weights $\lambda_{\text{recon}}, \lambda_{\text{KLD}}, \lambda_{\text{L2}}$, temperature $\tau$.
Initialize model parameters $\theta, \phi$.
**for** each training epoch **do**
    **for** each batch $\{(\boldsymbol{p}_j, \beta_j, \boldsymbol{x}_j)\}_{j=1}^{B} \subset \mathcal{D}$ **do**
        Compute activation: $\boldsymbol{a}_j \leftarrow \tau \cdot \text{sign}(\beta_j)/(|\beta_j| + \epsilon)$
        *// CVAE forward pass*
        Get posterior parameters: $(\boldsymbol{\mu}_{z_j}, \log \boldsymbol{\sigma}_{z_j}^2) \leftarrow \text{Encoder}(\boldsymbol{x}_j, \boldsymbol{p}_j; \theta)$
        Sample latent code via reparameterization: $\boldsymbol{z}_j \leftarrow \boldsymbol{\mu}_{z_j} + \boldsymbol{\sigma}_{z_j} \odot \boldsymbol{\epsilon}$ where $\boldsymbol{\epsilon} \sim \mathcal{N}(0, \mathbf{I})$
        Reconstruct input: $\hat{\boldsymbol{x}}_j \leftarrow \text{Decoder}(\boldsymbol{z}_j, \boldsymbol{p}_j; \theta)$
        *// Router forward pass*
        Get router probabilities: $\boldsymbol{\pi}_j \leftarrow \text{Router}(\boldsymbol{x}_j; \phi)$, where $\sum_{k=1}^{K} \pi_{j,k} = 1$
        *// Loss computation*
        Reconstruction loss: $\mathcal{L}_{\text{recon}} \leftarrow \frac{1}{B} \sum_{j=1}^{B} ||\boldsymbol{x}_j - \hat{\boldsymbol{x}}_j||_2^2$
        Router L2 regularization: $\mathcal{L}_{\text{L2}} \leftarrow \frac{1}{B} \sum_{j=1}^{B} ||\boldsymbol{\pi}_j||_2^2$
        Define activation-dependent prior mixture for each sample $j$:
        **for** $k = 1, \ldots, K$ **do**
            Prior mean: $\boldsymbol{\mu}_{p,k} \leftarrow \text{sign}(\boldsymbol{a}_j) \cdot (1 + k)$
            Prior variance: $\boldsymbol{\sigma}_p^2 \leftarrow |\boldsymbol{a}_j|$
        **end for**
        KL divergence loss:

$$\mathcal{L}_{\text{KLD}} \leftarrow \frac{1}{B} \sum_{j=1}^{B} \sum_{k=1}^{K} \pi_{j,k} \cdot D_{\text{KL}}\left(\mathcal{N}(\boldsymbol{\mu}_{z_j}, \text{diag}(\boldsymbol{\sigma}_{z_j}^2)) \,||\, \mathcal{N}(\boldsymbol{\mu}_{p,k}, \boldsymbol{\sigma}_p^2 \mathbf{I})\right)$$

        Total loss: $\mathcal{L} \leftarrow \lambda_{\text{recon}}\mathcal{L}_{\text{recon}} + \lambda_{\text{KLD}}\mathcal{L}_{\text{KLD}} + \lambda_{\text{L2}}\mathcal{L}_{\text{L2}}$
        Update parameters $\theta, \phi$ by descending the gradient $\nabla_{\theta,\phi}\mathcal{L}$.
    **end for**
**end for**

---

For the ROI-level analysis, which is equivalent to setting `condition=None`, the $\beta$ value is computed by averaging the individual $\beta_i$ values of all constituent voxels within the ROI.

## C  TRAINING DETAILS

| Hyperparameter | Voxel-level | ROI-level |
|---|---|---|
| *Model Architecture* | | |
| Latent Dimension ($z_{dim}$) | 256 | 256 |
| Position Embedding Dimension | 64 | N/A |
| Mixture Components ($K$) | 4 | 4 |
| *Optimization* | | |
| Batch Size | 128 | 128 |
| Epochs | 3 | 100 |
| Learning Rate | $1 \times 10^{-5}$ | $2 \times 10^{-4}$ |
| *Loss Function* | | |
| Reconstruction Weight ($\lambda_{\text{recon}}$) | 1.0 | 1.5 |
| KLD Weight ($\lambda_{\text{KLD}}$) | 1.0 | 1.0 |
| Router L2 Weight ($\lambda_{\text{L2}}$) | 20.0/400.0 | 20.0/400.0 |
| Temperature ($\tau$) | 1.0 | 1.0 |

Table C.1: Training hyperparameters for the voxel-level and ROI-level models. The voxel-level model is conditioned on spatial coordinates, whereas the ROI-level model is unconditional.

# D   DETAILS ON ENCODER AND DECODER IN CVAE

**Encoder.**   The encoder, parameterized by $\phi$, approximates the posterior distribution $q_\phi(\mathbf{z}|\mathbf{x}, \mathbf{p})$, where $\mathbf{z} \in \mathbb{R}^{D_z}$ is the latent variable. The positional vector $\mathbf{p}$ is first projected into a high-dimensional embedding $\mathbf{e}_p = \mathrm{NN}_{\mathrm{pos}}(\mathbf{p})$. This embedding is then concatenated with the CLIP feature vector $\mathbf{x}$ and processed by an MLP encoder to yield the parameters of the approximate posterior, which we assume to be a diagonal Gaussian:

$$q_\phi(\mathbf{z}|\mathbf{x}, \mathbf{p}) = \mathcal{N}(\mathbf{z}|\boldsymbol{\mu}_\phi(\mathbf{x}, \mathbf{p}), \mathrm{diag}(\boldsymbol{\sigma}_\phi^2(\mathbf{x}, \mathbf{p})))$$

The mean $\boldsymbol{\mu}_\phi$ and log-variance $\log \boldsymbol{\sigma}_\phi^2$ are computed as follows:

$$\mathbf{h}_{enc} = \mathrm{NN}_{\phi,enc}([\mathbf{x}; \mathbf{e}_p]) \tag{12}$$

$$\boldsymbol{\mu}_\phi = \mathbf{W}_\mu \mathbf{h}_{enc} + \mathbf{b}_\mu \tag{13}$$

$$\log \boldsymbol{\sigma}_\phi^2 = \mathbf{W}_{\sigma^2} \mathbf{h}_{enc} + \mathbf{b}_{\sigma^2} \tag{14}$$

where $[\cdot; \cdot]$ denotes concatenation, latent variable $\mathbf{z} = \boldsymbol{\mu}_\phi + \boldsymbol{\sigma}_\phi \odot \boldsymbol{\epsilon}$, with $\boldsymbol{\epsilon} \sim \mathcal{N}(\mathbf{0}, \mathbf{I})$.

**Decoder.**   The decoder, parameterized by $\theta$, reconstructs the CLIP feature vector from the latent sample $\mathbf{z}$ and the positional condition $\mathbf{p}$. It defines the likelihood distribution $p_\theta(\mathbf{x}|\mathbf{z}, \mathbf{p})$. The latent variable $\mathbf{z}$ is concatenated with the same positional embedding $\mathbf{e}_p$ and passed through the decoder MLP to produce the reconstruction $\hat{\mathbf{x}}$:

$$\mathbf{h}_{dec} = \mathrm{NN}_{\theta,dec}([\mathbf{z}; \mathbf{e}_p]) \tag{15}$$

$$\hat{\mathbf{x}} = \mathbf{W}_{out} \mathbf{h}_{dec} + \mathbf{b}_{out} \tag{16}$$

# E LINEAR AND NONLINEAR PROPERTIES OF THE BRAIN: HOW MANY VOXELS ARE SUFFICIENT FOR AN ROI?

For a given Region of Interest (ROI), assume it contains N voxels. Each voxel is associated with K semantics. For each voxel $i$, we take its semantic vector, $s_i$. These N principal semantic vectors are then concatenated to form a matrix $A = [s_1, s_2, \ldots, s_N]$, where $A \in \mathbb{R}^{d \times N}$. The number of linearly independent principal semantics within the ROI can be determined by computing the rank of matrix $A$ with a tolerance of $1 \times 10^{-5}$.

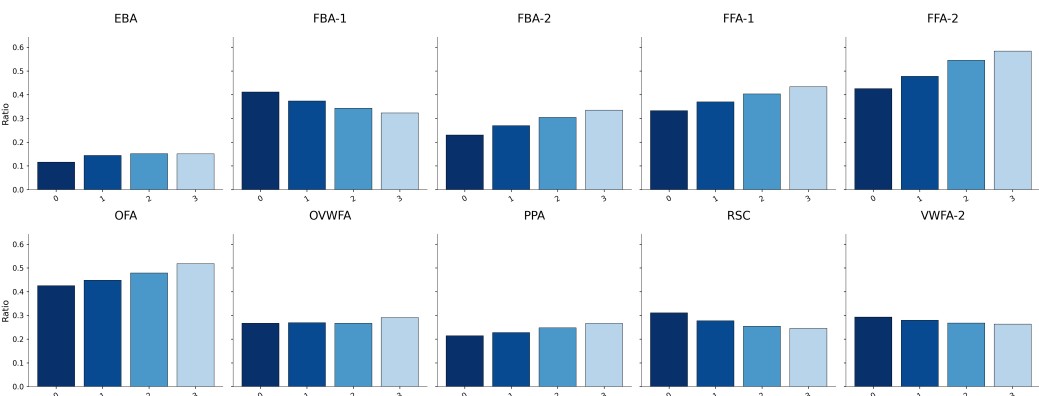

Figure E.1: **Normalized rank of the four semantic matrices for voxels across different ROIs.** The y-axis shows the matrix rank divided by the total number of voxels, representing the fraction of linearly independent semantic vectors.

Our results reveal that the normalized rank of almost all ROIs is below 60%, with some dropping to as low as 10%. This implies a high degree of linear dependency among voxel representations and suggests an inherent sparsity in the brain's coding scheme.

This, in turn, poses a fundamental question: How do the semantically distinct voxels effectively the basis vectors of this representational space interact? Furthermore, how do they leverage population coding to collectively represent complex information about the real world?

# F PROJECTOR (SEMANTIC → PREFIX)

We transform $\hat{\mathbf{x}}_k$ into a length-$n$ prefix sequence compatible with the LLM hidden space. The projector is a minimal MLP—linear transform, element-wise nonlinearity, and normalization—followed by a reshape:

$$\mathbf{v}_k^\flat = \mathrm{LN}\big(\sigma(W\,\hat{\mathbf{x}}_k + b)\big) \in \mathbb{R}^{nH}, \qquad \mathbf{V}_k = \mathrm{reshape}\big(\mathbf{v}_k^\flat, (n, H)\big) \in \mathbb{R}^{n \times H}, \qquad (17)$$

where $n$ is the prefix length, $H$ is the LLM hidden size, $\sigma(\cdot)$ is a pointwise nonlinearity (e.g., GELU), and $W, b$ are learnable parameters. The resulting $\mathbf{V}_k$ lies in the same embedding space as text and is used as a *visual prefix*. In practice, the prefix length $n$ is kept the same for training and inference, and the projector input dimension $d_c$ matches the semantic vector dimensionality produced by the VAE.

## G  REAL PHYSICAL PROXIMITY AND POSITION EMBEDDING ENCODED BY OUR MODEL

Our proposed method leverages the spatial coordinates of each voxel as conditional information for a CVAE. These coordinates are first transformed into a position embedding via an encoder, which is subsequently concatenated with both the input and the latent variable to guide the training process.

To visually assess the correspondence between the physical voxel space and the learned embedding space, we compared their topological structures using t-Distributed Stochastic Neighbor Embedding (t-SNE). We independently generated two 2D manifolds by applying t-SNE to: 1) the 3D physical coordinates, and 2) the $D_{pos}$ dim position embeddings. To facilitate a direct comparison, both manifolds are colored according to a unified proximity metric relative to a common reference voxel. The top plot, representing the physical space, is colored by the normalized Euclidean distance to the reference. The bottom plot, representing the embedding space, is colored by a normalized dissimilarity metric calculated as 1 - Cosine Similarity.(This ensures a consistent color map where darker regions represent closer proximity in both physical and embedding terms, allowing for a direct visual inspection of how well the embedding's structure preserves the original physical topology.) Our

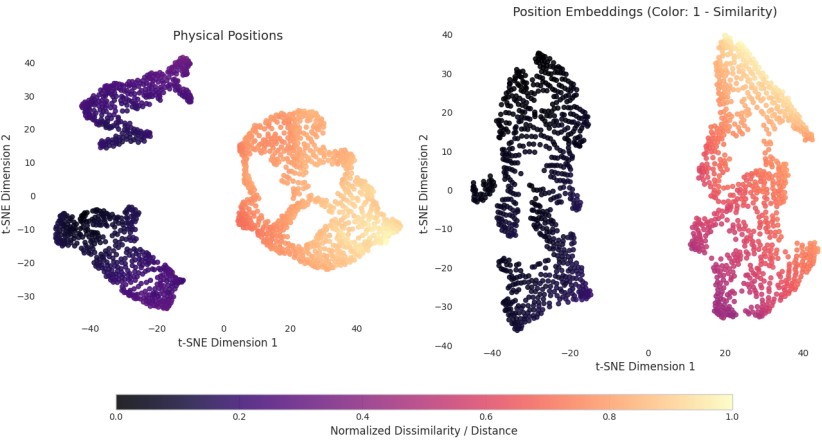

Figure G.1: OPA's t-SNE manifold picture

findings reveal a hierarchical organizational principle: at a macroscopic level, the semantic organization of voxels corresponds broadly to their physical layout, forming two distinct superclusters. At a more fine-grained, local level, however, semantic clustering diverges from strict physical adjacency. This suggests that while large-scale anatomy dictates the general semantic landscape, local voxel semantics are refined by factors beyond mere physical proximity.

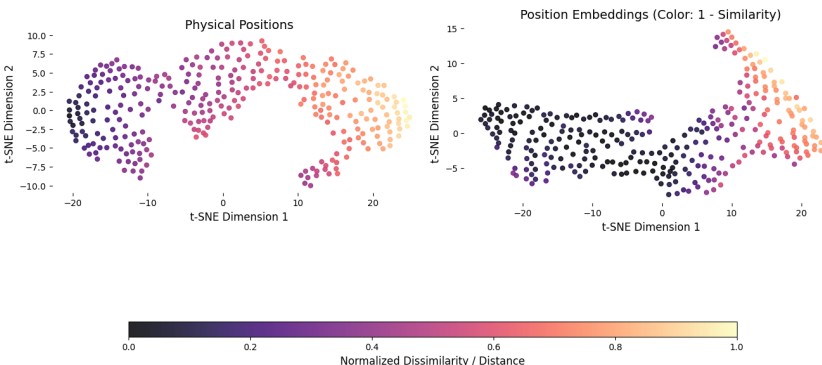

Figure G.2: FFA-2's t-SNE manifold picture

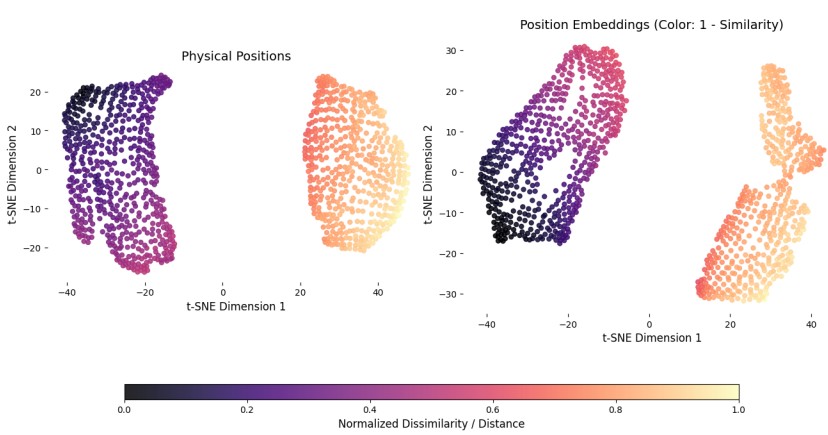

Figure G.3: PPA's t-SNE manifold picture

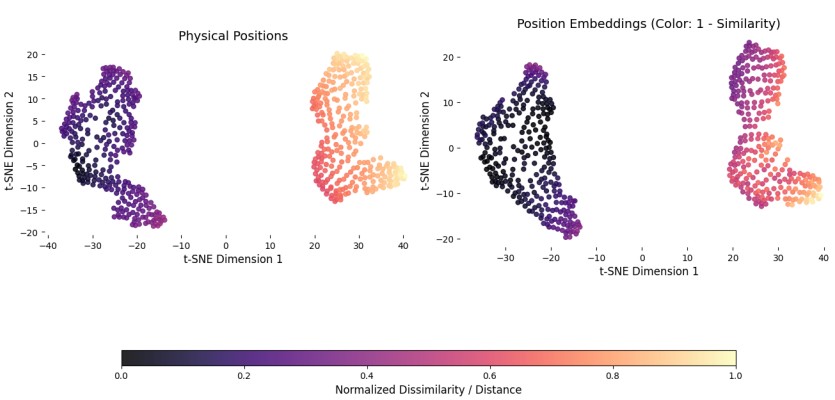

Figure G.4: RSC's t-SNE manifold picture

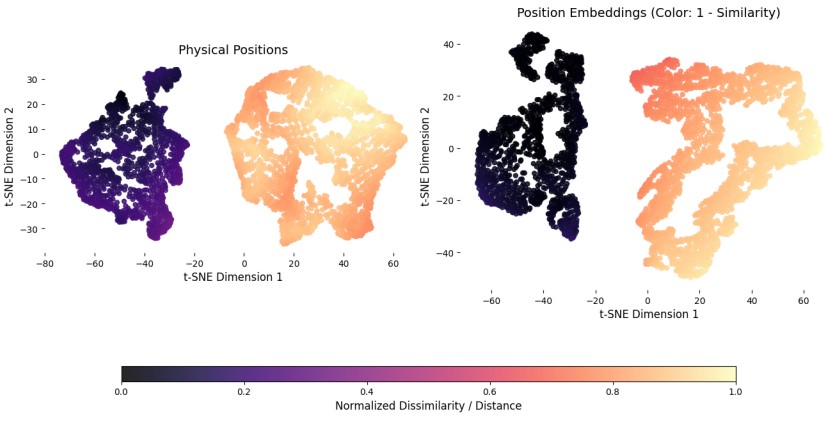

Figure G.5: EBA's t-SNE manifold picture

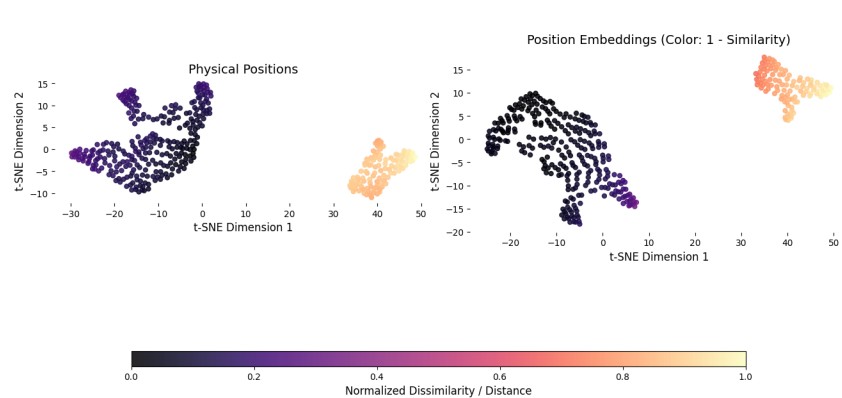

Figure G.6: FBA-1's t-SNE manifold picture

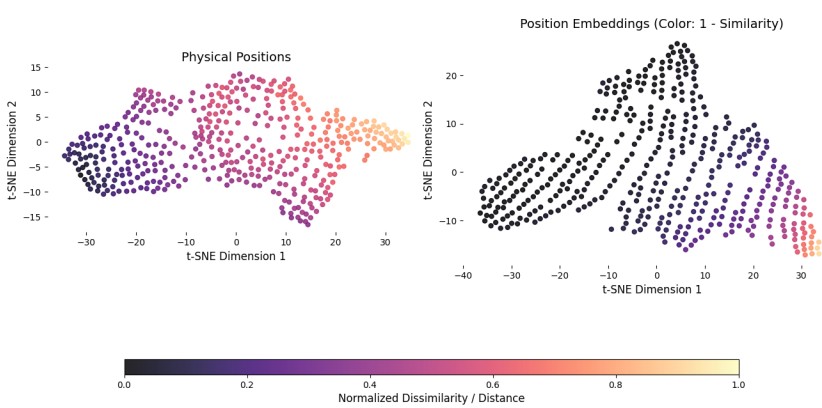

Figure G.7: FBA-2's t-SNE manifold picture

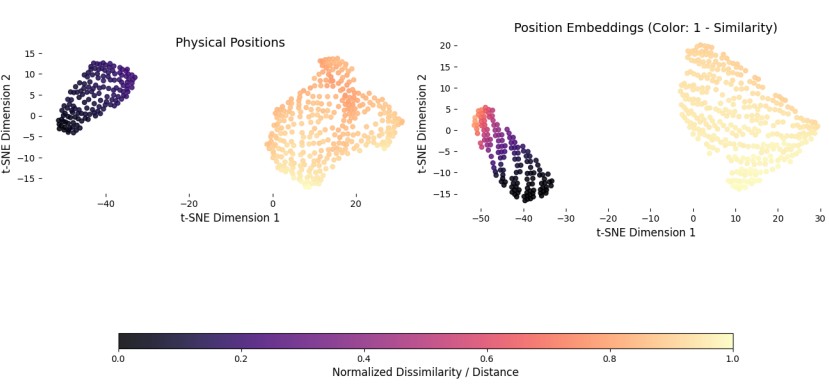

Figure G.8: OVWFA's t-SNE manifold picture

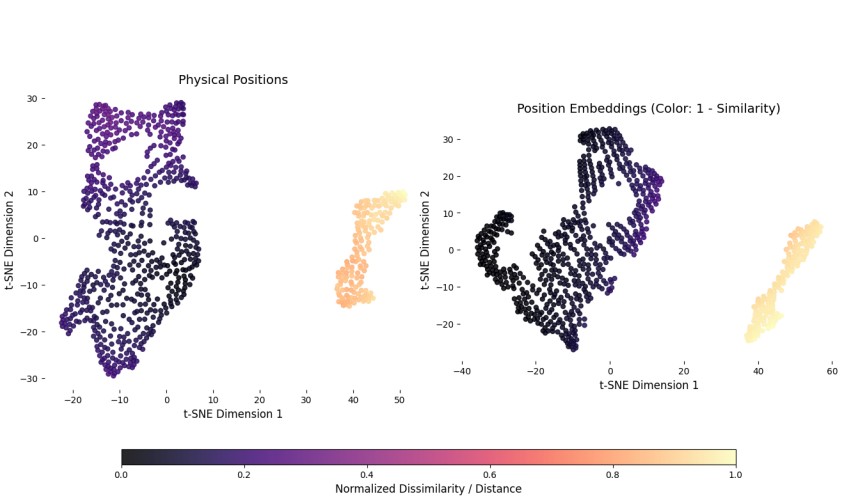

Figure G.9: VWFA-1's t-SNE manifold picture

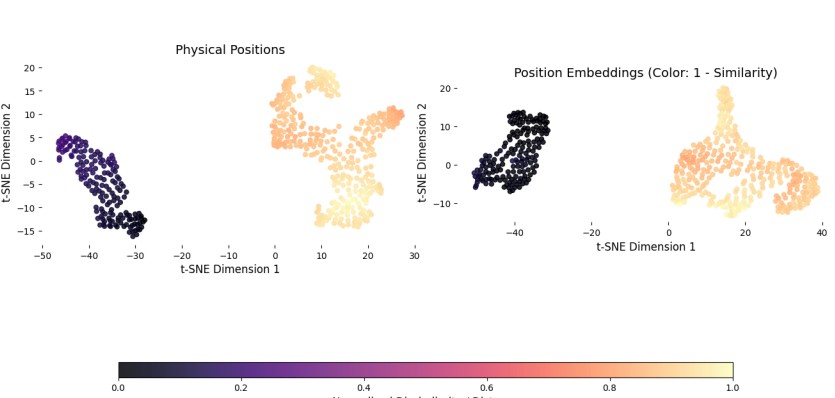

Figure G.10: VWFA-2's t-SNE manifold picture

# H EFFECT OF VOXEL PROXIMITY ON NEURAL DECODING CONSISTENCY

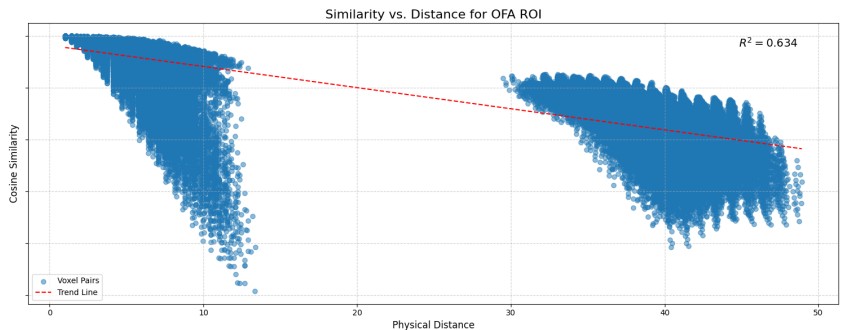

Figure H.1: Decoded Semantic Similarity vs. Physical Distance for Voxels within the OFAROI

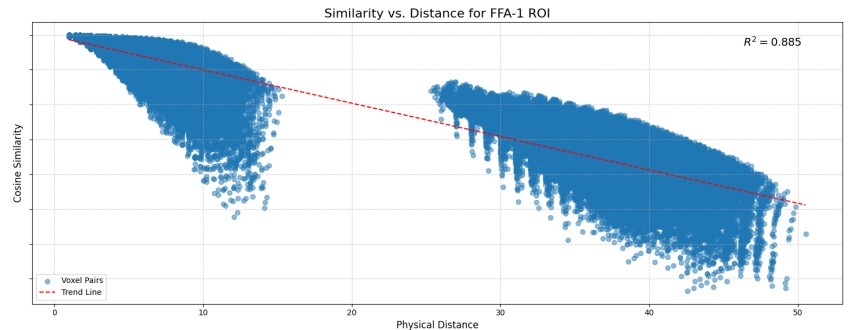

Figure H.2: Decoded Semantic Similarity vs. Physical Distance for Voxels within the FFA-1ROI

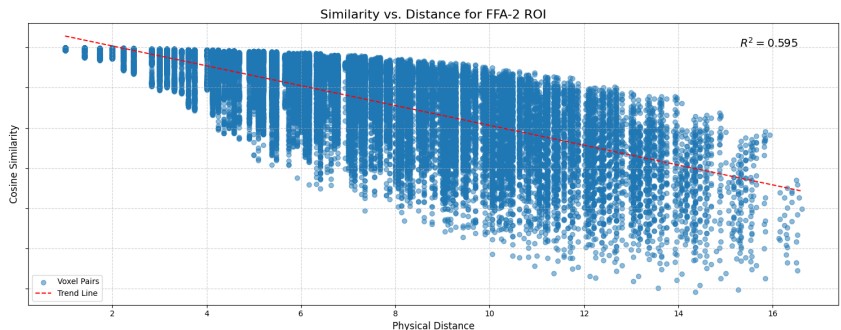

Figure H.3: Decoded Semantic Similarity vs. Physical Distance for Voxels within the FFA-2ROI

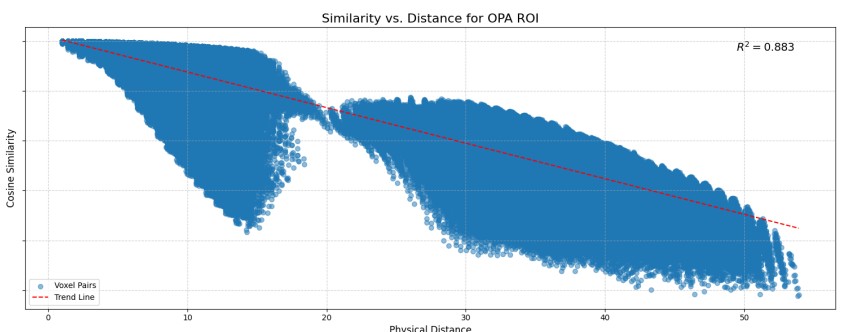

Figure H.4: Decoded Semantic Similarity vs. Physical Distance for Voxels within the OPAROI

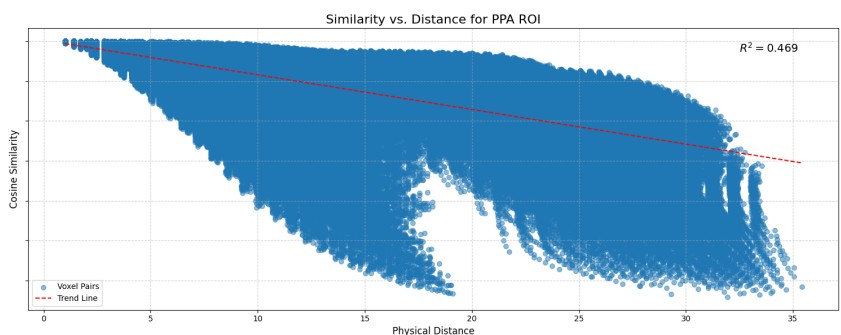

Figure H.5: Decoded Semantic Similarity vs. Physical Distance for Voxels within the PPAROI

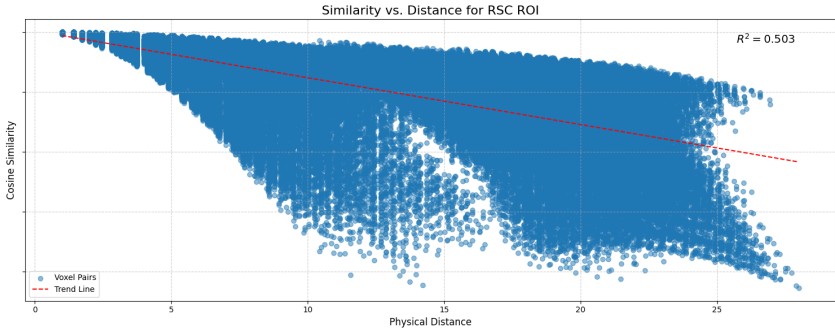

Figure H.6: Decoded Semantic Similarity vs. Physical Distance for Voxels within the RSCROI

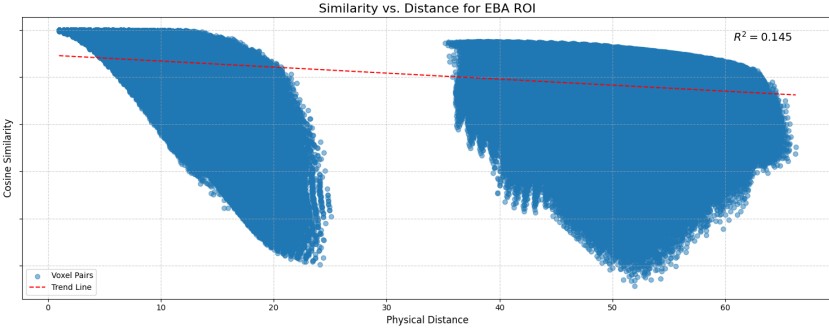

Figure H.7: Decoded Semantic Similarity vs. Physical Distance for Voxels within the EBAROI

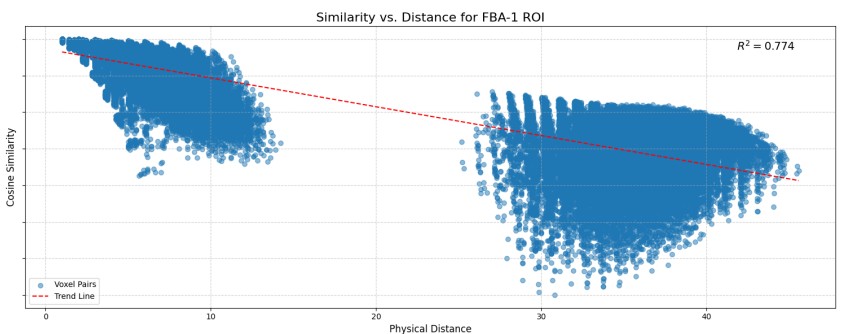

Figure H.8: Decoded Semantic Similarity vs. Physical Distance for Voxels within the FBA-1ROI

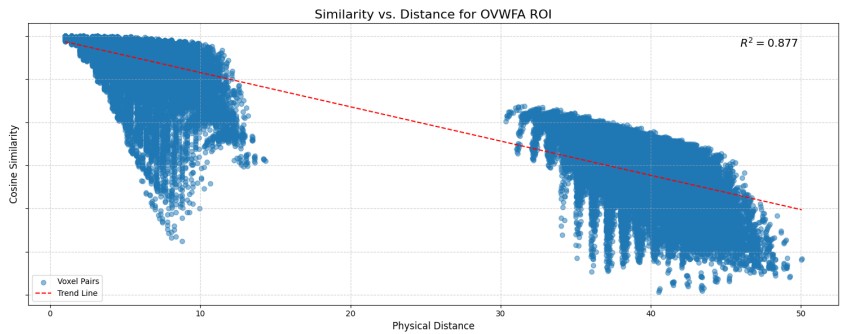

Figure H.9: Decoded Semantic Similarity vs. Physical Distance for Voxels within the OVWFAROI

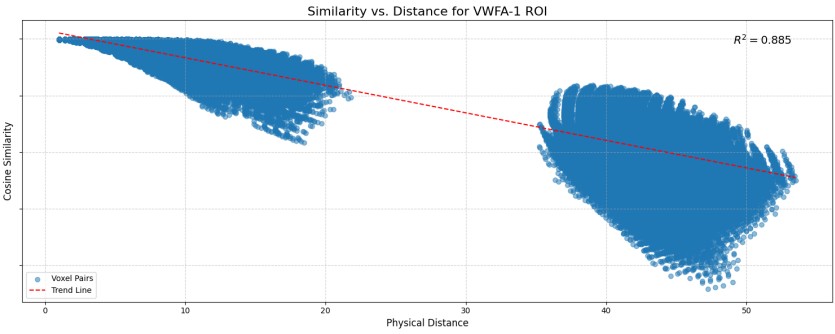

Figure H.10: Decoded Semantic Similarity vs. Physical Distance for Voxels within the VWFA-1ROI

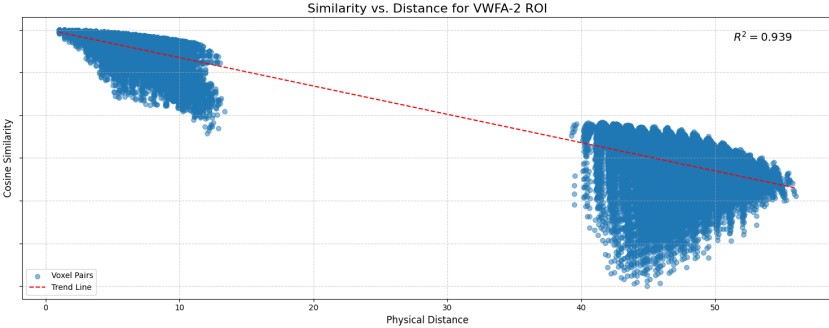

Figure H.11: Decoded Semantic Similarity vs. Physical Distance for Voxels within the VWFA-2ROI

## I  MULTI SELECTIVE

| Level | Region | ROI | Semantic 1 | Semantic 2 | Semantic 3 | Semantic 4 |
|---|---|---|---|---|---|---|
| **ROI-level** | Faces | OFA | 0.3350 | 0.4651 | 0.1999 | $4.55 \times 10^{-6}$ |
| | | FFA-1 | 0.5111 | 0.4889 | $2.29 \times 10^{-6}$ | $8.52 \times 10^{-7}$ |
| | | FFA-2 | 0.5151 | 0.4849 | $2.28 \times 10^{-6}$ | $1.09 \times 10^{-6}$ |
| | Places | OPA | 0.5128 | 0.4872 | $3.22 \times 10^{-6}$ | $1.10 \times 10^{-6}$ |
| | | PPA | 0.5119 | 0.4881 | $1.06 \times 10^{-5}$ | $1.46 \times 10^{-6}$ |
| | | RSC | 0.5929 | 0.4071 | $2.07 \times 10^{-6}$ | $7.61 \times 10^{-7}$ |
| | Bodies | EBA | 0.3663 | 0.4523 | 0.1813 | $5.75 \times 10^{-6}$ |
| | | FBA-1 | 0.5126 | 0.4874 | $2.38 \times 10^{-6}$ | $1.02 \times 10^{-6}$ |
| | | FBA-2 | 0.5133 | 0.4867 | $2.12 \times 10^{-6}$ | $8.48 \times 10^{-7}$ |
| | Words | OVWFA | 0.5199 | 0.4801 | $6.06 \times 10^{-6}$ | $1.22 \times 10^{-6}$ |
| | | VWFA-1 | 0.5161 | 0.4839 | $2.18 \times 10^{-6}$ | $8.41 \times 10^{-7}$ |
| | | VWFA-2 | 0.5161 | 0.4839 | $2.18 \times 10^{-6}$ | $8.41 \times 10^{-7}$ |
| **Voxel-level** | Faces | OFA | 0.5429 | 0.3767 | 0.0803 | $2.92 \times 10^{-10}$ |
| | | FFA-1 | 0.6314 | 0.3686 | $1.31 \times 10^{-9}$ | $8.75 \times 10^{-12}$ |
| | | FFA-2 | 0.6266 | 0.3734 | $1.18 \times 10^{-8}$ | $2.79 \times 10^{-10}$ |
| | Places | OPA | 0.5961 | 0.3819 | 0.0220 | $1.92 \times 10^{-12}$ |
| | | PPA | 0.5840 | 0.3847 | 0.0313 | $2.87 \times 10^{-12}$ |
| | | RSC | 0.6140 | 0.3842 | 0.0018 | $3.87 \times 10^{-11}$ |
| | Bodies | EBA | 0.5278 | 0.3705 | 0.1017 | $6.47 \times 10^{-11}$ |
| | | FBA-1 | 0.6321 | 0.3679 | $6.46 \times 10^{-10}$ | $4.10 \times 10^{-12}$ |
| | | FBA-2 | 0.6134 | 0.3866 | $7.61 \times 10^{-9}$ | $8.14 \times 10^{-12}$ |
| | Words | OVWFA | 0.5520 | 0.3852 | 0.0628 | $1.21 \times 10^{-11}$ |
| | | VWFA-1 | 0.5675 | 0.3908 | 0.0417 | $3.80 \times 10^{-12}$ |
| | | VWFA-2 | 0.6019 | 0.3805 | 0.0176 | $6.51 \times 10^{-12}$ |

## J  RESULTS OF VOXEL DECODING IN ALL ROI)

### J.1  FACES

### J.2  WORDS

| ROI | Voxel $(p_x, p_y, p_z)$ | Semantics |
|---|---|---|
| OFA | [18, 14, 39] | close cat face eyes; cat table cat food bowl; close cat bed; close cat couch |
| OFA | [18, 14, 40] | close cat face eyes; cat table cat; close cat bed; close cat couch |
| OFA | [18, 15, 39] | close cat face bird background; cat table cat; close cat bed; close cat couch |
| OFA | [18, 15, 40] | close up cat face bird background; cat table cat; close cat bed; close cat couch |
| OFA | [19, 13, 37] | close cat cage; cat table cat food bowl; close cat bed; close cat couch |
| OFA | [19, 13, 38] | close cat face ears; cat table laptop computer; close cat bed; close cat couch |
| OFA | [19, 15, 37] | close cat face bird background; cat table laptop computer; close cat bed; close cat couch |
| FFA-1 | [24, 21, 30] | zebra standing; zebra standing dirt ground; birds food beaks; people bird hands |
| FFA-1 | [24, 21, 31] | zebra stand zoo enclosure; zebra standing dirt ground; birds food beaks; people bird hands |
| FFA-1 | [24, 21, 32] | zebra standing; zebra standing zebra baby zebra mouth; birds food beaks; people bird hands |
| FFA-1 | [25, 17, 30] | zebra standing; zebra standing dirt ground; birds food beaks; people bird hands |
| FFA-1 | [25, 17, 31] | zebra stand zoo enclosure; zebra standing zebra baby zebra mouth; birds food beaks; people bird hands |
| FFA-1 | [25, 17, 32] | zebras; zebra standing dirt floor; birds food beaks; people bird hands |
| FFA-1 | [25, 18, 30] | woman baby kangaroo front fence; zebra standing dirt floor; birds food beaks; people bird hands |
| FFA-2 | [60, 38, 29] | man camera man shirt cat; men frisbee cell phone; people person baseball bat; people banana toothbrush |
| FFA-2 | [60, 38, 30] | men man camera; men frisbee cell phone; people frisbee; people banana toothbrush |
| FFA-2 | [60, 40, 30] | men man camera; men frisbee cell phone; people frisbee knife; people banana toothbrush |
| FFA-2 | [60, 41, 30] | man camera man wheelchair; men frisbee cell phone; people frisbee cell phone; people banana toothbrush |
| FFA-2 | [60, 42, 29] | man camera man wheelchair dog front fence; men frisbee cell phone; people frisbee knife; people plate food |
| FFA-2 | [60, 42, 30] | man camera man wheelchair dog front fence; men frisbee cell phone; people frisbee knife; people banana toothbrush |
| FFA-2 | [60, 42, 31] | man camera man wheelchair dog front fence; men frisbee cell phone; people frisbee cell phone; people plate hotdog bottle ketchup |
| aTL-faces | [25, 53, 21] | people kitchen laptop computer; people front laptop computer; people front television; televisions top table room window |
| aTL-faces | [25, 53, 22] | people kitchen laptop computer; people front laptop computer; people front television; televisions top table window |
| aTL-faces | [25, 53, 23] | people kitchen laptop computer; people front laptop computer; people front television; televisions top table window |
| aTL-faces | [25, 53, 24] | people kitchen laptop computer; people front laptop computer; people front television; televisions top table room window |
| aTL-faces | [25, 54, 24] | people kitchen laptop computer; people front laptop computer; people front television; televisions top table room window |

Table J.1: Example voxel-wise semantics for ROI = OFA. Each row lists voxel coordinates and the associated semantic descriptors.

| ROI | Voxel $(p_x, p_y, p_z)$ | Semantics |
|---|---|---|
| OVWFA | [22, 10, 39] | close up cell phone table; close up bird top desk; close up cat face cell phone; close up cat face cell phone |
| OVWFA | [22, 11, 34] | close up cell phone table; close up bird top desk; close up cat face cell phone; close up cat face cell phone |
| OVWFA | [22, 11, 35] | close up cell phone table; close up bird top table; close up cat face cell phone; close up cat face cell phone |
| OVWFA | [22, 11, 37] | close up cell phone table; close up bird top table; close up cat face cell phone; close up cat face cell phone |
| OVWFA | [22, 11, 38] | close up cell phone table; close up bird table; close up cat face cell phone screen; close up cat face cell phone |
| OVWFA | [22, 11, 39] | close up cell phone table; close up bird table; close up cat face cell phone screen; close up cat face cell phone screen |
| OVWFA | [22, 11, 40] | close up cell phone table; close up bird top table; close up cat face cell phone; close up cat face cell phone |
| OVWFA | [22, 12, 35] | close up cell phone table; close up bird top table; close up bird top desk; close up cat face cell phone |
| OVWFA | [22, 12, 36] | close up cell phone table; close up bird counter top; close up cat face cell phone screen; close up cat face cell phone |
| OVWFA | [22, 12, 37] | close up cell phone table; close up bird top table; close up cat face cell phone; close up cat face cell phone screen |
| VWFA-1 | [18, 26, 39] | close up food truck dashboard cup coffee tray truck; close plate food sandwich top tray; close plate food sandwich top table; plate food tray top table |
| VWFA-1 | [18, 26, 40] | close up food truck dashboard cup coffee tray; close plate food sandwich top tray; close plate food knife side; plate food tray top table |
| VWFA-1 | [18, 27, 39] | close up food truck dashboard cup coffee tray truck; close plate food sandwich tray; close plate food sandwich top table; close plate food knife |
| VWFA-1 | [18, 27, 40] | close up food truck dashboard cup coffee tray truck; close plate food sandwich tray; close plate food sandwich fries top table; close plate food knife |
| VWFA-1 | [19, 15, 34] | close up sandwich tray tray; close plate food sandwich fries top; close plate food sandwich fries top table; plate food tray top table |
| VWFA-1 | [19, 15, 35] | close up sandwich tray tray; close plate food sandwich fries; close plate food knife side; plate food tray top picnic table |
| VWFA-1 | [19, 15, 36] | close up sandwich tray tray tray tray tray t; close plate food sandwich fries; slices toast sandwich top; close sandwich plate beverage side |
| VWFA-1 | [19, 15, 37] | close up sandwich tray tray tray tray tray t; close plate food sandwich fries; slices toast sandwich top; plate food tray top picnic table |
| VWFA-1 | [19, 16, 33] | close up sandwich tray tray; close plate food sandwich fries; close sandwich plate beverage side; close plate food knife top table |

Table J.2: Voxel-wise semantics for ROI = OVWFA. Each row lists voxel coordinates and associated semantic descriptors.