# OpenReview forum: "BrainMIND: Interpret Fine-grained Spatial Mapping of Brain Activity to Multi-semantic Concepts"
_ICLR.cc/2026/Conference — ICLR 2026 Conference Withdrawn Submission_

### Official Review · Reviewer_oNbE · 2025-10-15

**Soundness:** 2
**Presentation:** 1
**Contribution:** 2
**Rating:** 2
**Confidence:** 4

**Summary:**

The paper proposes BrainMIND, a conditional variational autoencoder (CVAE) with a dynamic mixture-of-Gaussians prior, conditioned on both fMRI voxel positions and brain responses, to decode multi-semantic representations from the Natural Scenes Dataset (NSD). The decoded latent features are mapped into CLIP space and then converted to natural language via a fine-tuned large language model (LLM). The authors claim this approach reveals mixed selectivity at both the region and voxel levels and provides interpretable multi-semantic mappings across the visual cortex.

**Strengths:**

-The work addresses a meaningful and underexplored problem — interpretable, voxel-level, multi-semantic decoding rather than simple reconstruction or region-level prediction.

-Authors connect findings to established cortical selectivity patterns (FFA, PPA, VWFA) and report consistency with known priors, suggesting neuroscientific validity.

-The paper promises open code and weights, and the ethics section is solid and transparent.

**Weaknesses:**

generally the text is poorly written, with some mistakes and lot of AI generated content. This is not a problem per-se but please double check the whole text and especially the citations. Many of them are kind of wrong with maybe the right author but incorrect titles or the other way around.

Furthermore, the paper oscillates between semantic reconstruction and interpretability. It’s unclear whether BrainMIND is intended as a decoding model (predicting content from brain data) or as a representational analysis tool. If it is intended as a decoding model lot of comparison is missing with existent literature. There is only one comparison with BrainSCUBA, with limited improvements.

The paper completely lack evaluation, do we need the router? What is the impact?

Also I'm not fully convinced by the math presented in the paper. For example, the gating is not-differentiable, but there are other potential flaws.

**Questions:**

Honestly I think the paper should be improved in several directions:

- Improved text for clarity and readability
- Improved explanation of what is the research question here, and why (and how) the method proposed is solving the question
- Clear math
- Fair comparison with prior work (properly cited)

---

> ### Author Response · Authors · 2025-11-13
> **Apology for incorrect references and thanks for valuable feedbacks**
>
> Thank you very much for taking the time to review our paper and pointing out the issues. We sincerely apologize for the incorrect literature references. We have corrected the references and included several necessary reproducibility details in the updated PDF. And we want to confirm that the main body of the research work is entirely based on authentic and reproducible work, without any fabricated data, and we will withdraw our paper and offer our sincere apologies for this oversight.
>
> Meanwhile, we thank the reviewer for their valuable feedbacks. Our definition of "decoding" is to identify and decode inputs that mostly activate a given voxel, we highlight the "voxel-wise decoding" of a single voxel's most selective inputs, compared to reconstructing the inputs and decoding from population of voxels, we have modified into "selectivity" as a better phrase to avoid confusion. We will also include comparisons with other relevant baselines for decoding and representational analysis tool in our future work as suggested by the reviewer. Our router is to assign each voxel with different concepts based on the inputs and their physical locations, to generate multi-semantic concepts associated with each voxel. We will include more ablation studies, and we have revised the math part for better clarity. Our framework is guaranteed to be differentiable. We apologize for any confusion about the original math formulas.

---

### Official Review · Reviewer_L1We · 2025-10-24

**Soundness:** 2
**Presentation:** 2
**Contribution:** 2
**Rating:** 2
**Confidence:** 5

**Summary:**

The paper proposes BrainMIND, a voxel-level fMRI decoding framework for interpretable brain-to-text reconstruction. Voxel-wise neural decoding is an important problem for understanding the relationship between human brain representations and latent embeddings of intelligent models. However, the manuscript contains several serious issues that undermine confidence in the work.

**Strengths:**

Exploring fine-grained mapping between brain activity patterns and AI model representations is a meaningful direction in bridging neuroscience and AI.

**Weaknesses:**

* The bibliography contains multiple citation errors. For example, several references list non-existent or placeholder author names such as [1–2], and some cited works cannot be traced to public sources or published papers (e.g., [2]).

  [1] Aoxiao Luo, John D Smith, and Jane Doe. Brain diffusion for visual exploration: Cortical discovery using large scale generative models. arXiv preprint arXiv:2306.03089, 2023.

  [2] Yujia Wang, John D Smith, and Jane Doe. Incorporating clip into brain decoding: Zero-shot learning for fmri analysis. NeuroImage, 250:118956, 2022.

* Another major concern to me is the lack of comparison to existing work on the topic of fMRI-to-caption decoding. While the paper compares against BrainSCUBA, the brain-to-text decoding field already contains numerous advanced methods such as [1-3].

  [1] Neuro-Vision to Language: Enhancing Brain Recording-based Visual Reconstruction and Language Interaction. NeurIPS 2024.

  [2] Exploring the Visual Feature Space for Multimodal Neural Decoding. ICCV 2025.

  [3] Bridging the Gap between Brain and Machine in Interpreting Visual Semantics: Towards Self-adaptive Brain-to-Text Decoding, ICCV2025.

* The current methods lack crucial details for understanding.

**Questions:**

Implementation details are missing, such as the description and configuration of comparison methods, the choice of LLM, raising questions about reproducibility.

---

> ### Author Response · Authors · 2025-11-13
> **Apology for incorrect references and thanks for valuable feedbacks**
>
> Thank you very much for taking the time to review our paper and pointing out the issues. We sincerely apologize for the incorrect literature references. We have corrected the references and included several necessary reproducibility details in the updated PDF. And we want to confirm that the main body of the research work is entirely based on authentic and reproducible work, without any fabricated data, and we will withdraw our paper and offer our sincere apologies for this oversight. Meanwhile, we thank the reviewer for the valuable feedbacks for providing additional baselines for further comparisons, we will include these updates in our future work.

---

### Official Review · Reviewer_p5of · 2025-10-30

**Soundness:** 1
**Presentation:** 1
**Contribution:** 1
**Rating:** 0
**Confidence:** 5

**Summary:**

This paper proposes BrainMIND, a position‑conditioned CVAE with a dynamic mixture prior that decodes semantics embeddings and uses LLMs to produce natural‑language semantics at both ROI and voxel levels. I found that  the submission raises serious Ethical & Reproducibility Concerns. There are multiple placeholder‑style or internally inconsistent references. These issues impede a fair scientific assessment at this time.

**Strengths:**

n.a.

**Weaknesses:**

n.a.

**Questions:**

n.a.

**Details Of Ethics Concerns:**

The References contain repeated placeholder‑style author names, across multiple entries—e.g., Ferrante et al., 2023 (p. 10), Luo et al., 2023 (p. 11), Wang et al., 2022 (p. 12). There is also an underspecified citation “Clip‑decoding … In NeurIPS 2023” with minimal metadata (p. 10), which I cannot find the original one. There are also instances where completely unrelated papers are miscited. For example, “Seeing through things: Exploring the design space of privacy‑aware data‑enabled objects.” (p. 10) appears misaligned with the brain‑decoding topic and is likely mis‑cited within this context.
Such patterns are red flags for reference accuracy and raise concerns about automated/AI‑assisted generation. Moreover, these issues undermine the credibility of the manuscript and its experiments, and the inclusion of non‑existent citations makes peer review difficult.

---

> ### Author Response · Authors · 2025-11-13
> **Apology for incorrect references and clarification for authenticity of main research content**
>
> Thank you very much for taking the time to review our paper and pointing out the issues. We sincerely apologize for the incorrect literature references. We have corrected the references and included several necessary reproducibility details in the updated PDF. And we want to confirm that the main body of the research work, including the idea, methodology, experiments, and reported results, is entirely based on authentic and reproducible work, without any fabricated data. The full source code can be provided for further verification and reproducibility. We will withdraw our paper and offer our sincere apologies for this oversight.

---

### Official Review · Reviewer_txyh · 2025-11-01

**Soundness:** 1
**Presentation:** 1
**Contribution:** 1
**Rating:** 0
**Confidence:** 5

**Summary:**

This paper proposes a way to generate words from voxel-wise selectivity. The task is not new, and this paper contains numerous issues with AI generated citations.

**Strengths:**

Investigating the selectivity in higher visual cortex is an interesting problem.

**Weaknesses:**

I have serious serious concerns regarding this paper.

There are numerous citations which are totally made up and reference non-existent papers:
* Yujia Wang, John D Smith, and Jane Doe. Incorporating clip into brain decoding: Zero-shot learning for fmri analysis. NeuroImage, 250:118956, 2022.
* Aoxiao Luo, John D Smith, and Jane Doe. Brain diffusion for visual exploration: Cortical discovery using large scale generative models. arXiv preprint arXiv:2306.03089, 2023.
* Enrico Ferrante, John D Smith, and Jane Doe. Brain captioning: Decoding human brain activity into images and text. arXiv preprint arXiv:2305.11560, 2023.
* Justin Giles, Andrew Luo, and Leyla Isik. Clip-decoding: A generalist brain decoder for reconstructing arbitrary image-caption pairs. In Thirty-seventh Conference on Neural Information Processing Systems, 2023.


Second, by the commonly accepted definitions of encoding and decoding by Thomas Naselaris, this work does not perform decoding, yet says `Our framework achieves multi-semantic and position-aware decoding at both the coarse ROI scale and the fine-grained voxel level`. This work is **investigating selectivity** not decoding. This sentence shows that the authors are seemingly unaware of the question they are investigating.

Third, the figures are very strange. There are no cortical flat-maps or inflated maps, instead the authors seemingly plot out all figures using voxel positions. Which may change depending on the reference frame the MRI was transformed into. Note that this may vary from subject to subject.

Fourth, the authors at no point in the paper describe how the words for each voxel are generated.

**Questions:**

N/A

**Details Of Ethics Concerns:**

Numerous AI generated citations.

---

> ### Author Response · Authors · 2025-11-13
> **Apology for incorrect references and thanks for valuable feedbacks**
>
> Thank you very much for taking the time to review our paper and pointing out the issues. We sincerely apologize for the incorrect literature references. We have corrected the references and included several necessary reproducibility details in the updated PDF. And we want to confirm that the main body of the research work is entirely based on authentic and reproducible work, without any fabricated data. We will withdraw our paper and offer our sincere apologies for this oversight.
>
> Meanwhile, we are grateful for these constructive feedbacks from the reviewer. Our definition of decoding is to identify and decode inputs that mostly activate a given voxel, we highlight the "voxel-wise" decoding of single voxel's most selective inputs compared to decoding from population of voxels and reconstructing the inputs. We thank the suggestion of using "selectivity" as a better phrase, we have updated our revision. Meanwhile, we also agree that surface-based representations such as cortical flat-maps or inflated surfaces would provide clearer anatomical interpretability. We will update these suggestions in our revision. To clarify how words for each voxel are generated, we first associate each voxel with a set of semantic embedding vectors that produces maximal activations. These top-activating embeddings are then fed into a pretrained language model, which translates the embedding content into human-interpretable words or short textual descriptions.

---

### Author Response · Authors · 2025-11-13
**Apology for incorrect references and clarification for authenticity of main research content**

Dear PC, AC, and Reviewers,

We sincerely apologize for the incorrect literature references, as disclosed in the check list, we only intend to use LLMs to assist and polish writing, while we improperly use these tools in searching and formatting a few references. We take full responsibility for not carefully validating the sources of their references.  We have updated the PDF accordingly, with incorrect references replaced, and also incorporated additional feedbacks as reviewers suggested. Meanwhile, we want to confirm that the main body of the research work, including the idea, methodology, experiments, and reported results, is entirely based on authentic and reproducible work, without any fabricated data. The full source code can be provided for further verification and reproducibility. We will withdraw our paper and offer our sincere apologies for this oversight. We will take greater care and will strictly follow any submission guidelines in our future work.

---

> ### Comment · Reviewer_txyh · 2025-11-13
> **Still incorrect citations**
>
> Even in the current updated version, the second reference that I check is incorrect and appear hallucinated.
>
> 1. MIchael J. Tarr Leila Wehbe Andrew F. Luo, Margaret M. Henderson. Brainscuba: Fine-grained natural language captions of visual cortex selectivity. -> Author order is wrong
>
> I see another two references that are wrong, that I will not list here. I leave this as a challenge to the authors to identify the other references. I hope the authors can take ICLR seriously, and avoid AI generated citations.

---

> > ### Author Response · Authors · 2025-11-13
> >
> > Thank you again for your feedback and understanding. We take ICLR seriously and sincerely apologize for our oversight. The remaining errors were introduced during our manual proofreading last time. We have been careful not to use any AI-generated references in the last revision. Yet unfortunately, we failed to notice and correct the ordering issue in this example. We now use the BibTeX formatting for all references, and the PDF has been updated accordingly.

---

### Note · Authors · 2025-11-17

I have read and agree with the venue's withdrawal policy on behalf of myself and my co-authors.